# Local read haplotagging enables accurate long-read small variant calling

Alexey Kolesnikov[1], Daniel Cook[1], Maria Nattestad[1], Lucas Brambrink[1], Brandy McNulty[2], John Gorzynski [3], Sneha Goenka[3], Euan A. Ashley [3], Miten Jain [4], Karen H. Miga [2], Benedict Paten [2], Pi-Chuan Chang [1], Andrew Carroll [1] ✉ & Kishwar Shafin[1] ✉

Long-read sequencing technology has enabled variant detection in difficult-to-map regions of the genome and enabled rapid genetic diagnosis in clinical settings. Rapidly evolving third-generation sequencing platforms like Pacific Biosciences (PacBio) and Oxford Nanopore Technologies (ONT) are introducing newer platforms and data types. It has been demonstrated that variant calling methods based on deep neural networks can use local haplotyping information with long-reads to improve the genotyping accuracy. However, using local haplotype information creates an overhead as variant calling needs to be performed multiple times which ultimately makes it difficult to extend to new data types and platforms as they get introduced. In this work, we have developed a local haplotype approximate method that enables state-of-the-art variant calling performance with multiple sequencing platforms including PacBio Revio system, ONT R10.4 simplex and duplex data. This addition of local haplotype approximation simplifies long-read variant calling with DeepVariant.

Long-read sequencing technology can reach low-mappability regions where short-reads have difficulty in mapping correctly[1–5]. Long-read sequencing has helped to generate highly contiguous genome assemblies[6–8], extend variant benchmarking to complex regions[9,10] and has improved the quality and completeness of reference genomes[11–14].

Variant detection with long-reads takes advantage of the high mappability of long reads to reach difficult-to-map regions of the genome[1,15,16]. Long-read sequencing technology have lower base-level accuracy compared to short read technologies[1,17–20]. However, variant callers based on deep neural networks (DNN) can produce highly accurate variant calls with long-reads[16,21,22]. Recently, high-throughput long-read sequencing paired with accurate variant calling has demonstrated fastest genetic diagnosis in clinical setting[23–25]. Also, accurate long-reads show a higher diagnosis rate among individuals who were previously undiagnosed[26,27]. Long-read sequencing paired with methods based on machine learning for variant detection shows promise to have a far reaching impact in healthcare[28].

DNN-based variant callers not only take advantage of the mappability of the long reads, it also uses local haplotyping information to inform the neural network for accurate genotyping[16,21]. Previously[29], we have shown that using local haplotype information to determine the genotype of a variant improves the accuracy of variant detection with PacBio long-reads. In this three-step approach, a first round of variant detection is performed using DeepVariant[30] without haplotag information in the reads. Then, WhatsHap[31] is used to haplotag the reads. Finally, we run DeepVariant with the haplotag information to produce higher quality variants. A similar approach was taken for PEPPER-Margin-DeepVariant[16] pipeline for nanopore long-reads where Margin is used to haplotag the reads.

Other Long-read germline variant callers such as Clair3[21] and Medaka[32] use local haplotagging information from haplotagging methods like WhatsHap[31], Margin[33] or LongPhase[34]. Although the previously described three-step approach for variant detection provides accurate variants, it is difficult to tune three separate methods for each

[1]Google Inc, 1600 Amphitheatre Pkwy, Mountain View, CA, USA. [2]UC Santa Cruz Genomics Institute, University of California, Santa Cruz, CA, USA. [3]Stanford University, Stanford, CA, USA. [4]Northeastern university, Boston, MA, USA. ✉e-mail: awcarroll@google.com; shafin@google.com

new datatype or platform introduced for long-reads. The long-read platforms like Pacific Biosciences (PacBio), and Oxford Nanopore Technologies (ONT) are rapidly evolving[1,35–37] and a simpler variant calling approach that can enable accurate variant calling across different platforms is desirable.

Pacific Biosciences (PacBio) is a single-molecule real-time (SMRT) sequencing platform that uses circular consensus sequencing[29]. Recently, PacBio introduced a high-throughput machine called Revio[38] that uses transformer-based consensus sequence correction method DeepConsensus[39] on the instrument to generate highly accurate (99.9%) reads with length between 15kb and 20kb. The PacBio Revio machine can generate 30x human genome with one SMRT cell compared to 8x per SMRT cell with the previous Sequel-II machine which lowers the overall cost and turnaround time[40–42].

Oxford Nanopore Technologies (ONT) introduced R10.4 chemistry with two read types simplex and duplex[43]. Compared to R9.4.1 chemistry, R10.4 provides better resolution for homopolymer detection[43,44]. The simplex sequencing mode shows average read quality of 98% and duplex reads with 99.9% accuracy[44]. The nanopore long reads can be 100kb+ which makes it suitable for high-quality assemblies[7].

In this work, we introduce an approximate haplotagging method that can locally haplotag long reads without having to generate variant calls. Our approach uses local candidates to haplotag the reads and then the deep neural network model uses the haplotag approximation to generate high-quality variants. This approach eliminates the requirement for having the first two steps for haplotagging the reads and reduces the overhead for extending support to newer platforms and chemistries. We show that approximate haplotagging with candidate variants has comparable accuracy to haplotagging with WhatsHap. We report comparable or higher variant calling accuracy compared to DeepVariant-WhatsHap-DeepVariant approach with PacBio HiFi data. We demonstrate that extending support to the PacBio Revio machine can achieve high INDEL and SNP accuracy (SNP F1-score of 0.9993 and INDEL F1-score of 0.993). We also report support for ONT R10.4 simplex and duplex dataset with requiring PEPPER-Margin upstream of DeepVariant. We demonstrate INDEL F1-score of 0.84 and SNP F1-score of 0.9976 for nanopore simplex data and INDEL F1-socre of 0.90 and SNP F1-score of 0.999 for duplex data which is to our knowledge the highest variant calling accuracy achieved with nanopore long-reads.

## Results
### Local haplotype approximation method for genotyping with DeepVariant
DeepVariant performs variant calling in three stages: `make_examples`, `call_variants` and `postprocess_variants`. Previously, we have described each stage in detail[16,29,30]. Here, we briefly summarize each stage:

1. `make_examples`: This stage identifies candidate variants and creates a multichannel tensor representation (also known as an "example") for each candidate based on the read pileup at the candidate position. Each example encodes features such as read bases, read quality, mapping quality, read support for candidates, and read strands.
2. `call_variants`: Examples are then given to a Convolutional Neural Network (CNN) that determines the genotype likelihood for the candidate variant represented in each example.
3. `postprocess_variants`: Finally, we take the likelihoods from the CNN and report variants with their assigned genotype.Achieving accurate variant calling using long-read sequencing data requires calling variants twice often in a three-step process[16,29] (Fig. 1a, DeepVariant-WhatsHap-DeepVariant):
1. **First round variant calling**: First, we run DeepVariant on non-haplotagged alignment file to derive a set of variants.

2. **Haplotagging**: Using the variants from the non-haplotagged alignment file a secondary method WhatsHap is applied to haplotag the reads.
3. **Second round variant calling**: Finally, the haplotagged read is given to DeepVariant where DeepVariant uses extra features as haplotype channel to generate higher-quality variant calls.

The multi-step variant calling process of DeepVariant-WhatsHap-DeepVariant (Fig. 1a) introduces overhead to extend support to newer long-read data types as multiple steps need to be optimized. In this work, we implement an approximate local haplotagging method within DeepVariant to locally haplotag the reads to avoid having to run an external haplotagging method or to have to reperform variant calling following initial haplotagging (Fig. 1b). A description of the local haplotagging method is described here.

DeepVariant processes input by dividing it into 25,000-base pair windows and processes the windows in parallel to create examples[30]. The haplotagging algorithm is integrated into the `make_examples` step, utilizing raw candidate data to calculate local haplotagging for each 25,000-base pair window. For each window, DeepVariant generates candidate variants by identifying positions that differ from the reference genome. The haplotagging process ensures that only potential heterozygous SNP candidates are selected. This involves excluding all candidates that have only one alternate allele, and the reference allele is supported by fewer than three reads. Additionally, any candidates featuring alternate alleles that are not SNPs are also eliminated. Reads are then dynamically updated with haplotagging information. The haplotagging algorithm employs dynamic programming to determine the best haplotagging score for each possible haplotag assignment at each variant position within the window. A set of reads that overlap both the previous putative heterozygous variant location and the target location are used to calculate this score. The optimal allele haplotagging is determined by backtracking from the best score for the last variant position. After all alleles are assigned haplotags, read haplotags are assigned based on the majority of alleles that the read overlaps.

We use a graph to store read-support information, where each vertex represents an allele (either an alternate allele or the reference allele). This graph is constructed from candidates identified within a 25,000-base pair window. Only candidates that have sufficient evidence to be considered heterozygous and contain only SNP alleles are used. At each position, vertices are created for each SNP allele. Additionally, a vertex is created for the reference allele if there is adequate read support for it. It is possible to create more than two vertices at some positions if there is substantial support for both the reference and multiple alternate alleles. Edges are then created to connect consecutive vertices if a read overlaps these alleles.

We assume that the score representing each possible haplotagging at a given position can be extended to the next position. Haplotagging is performed locally for every 25,000-base pair window. The score is calculated for each possible haplotagging at each genomic position where putative heterozygous candidates are created. The score is extended from the first position to the last. If there are no incoming edges for any of the vertices representing a genomic position, haplotagging cannot be extended, and we must reinitialize the scores. In cases where one of the vertices at a genomic position lacks incoming edges, yet a path exists from other vertices at the same position, artificial edges are created. These artificial edges fully connect the vertex with all preceding vertices.

Here is a brief step-by-step description of the algorithm:
1. Graph is built where each vertex is an allele (alt or ref). Edges are created to connect consecutive vertices if there is a read overlapping these alleles.
2. Initial scores are calculated for all possible haplotagging for the first genomic position.

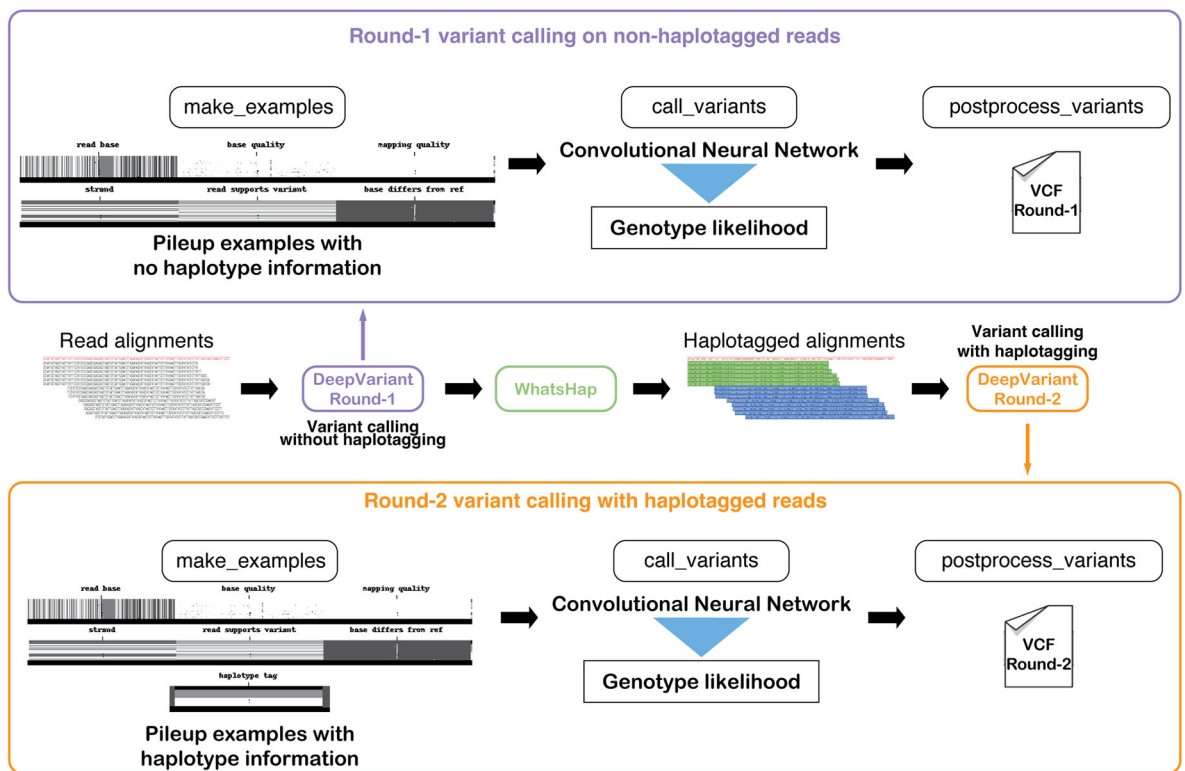

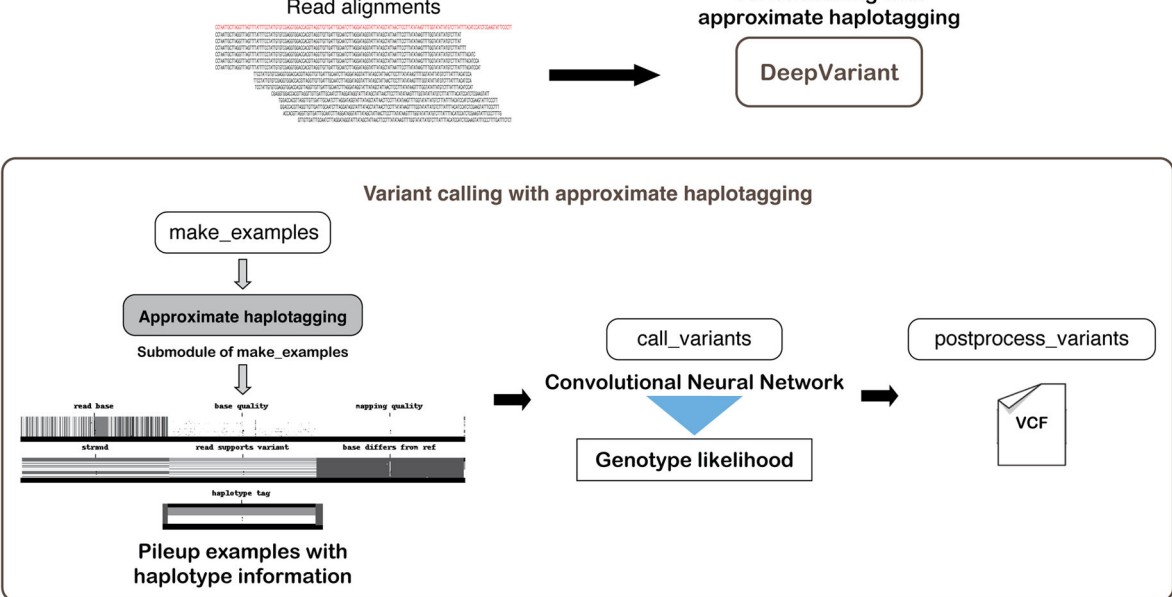

**Fig. 1 | Illustration of different long-read variant calling schema. a** Two-step variant calling process that uses WhatsHap to haplotag the reads. **b** Simplified variant calling process schema with approximate haplotagging method implemented within DeepVariant.

3. Scores are calculated at each position using best scores calculated for previous vertices.
4. The best score is backtracked from the last position. Each pair of alleles are assigned haplotags.
5. Reads are assigned haplotags based on a set of alleles reads overlap.

The outcome of the local approximate haplotagging is a set of reads with haplotype association of haplotype-1, haplotype-2 or non-haplotagged reads for each 25kb chunk. Then for each candidate variant observed in the chunk, we generate examples through the `make_examples` step of DeepVariant that creates a multi-channel

representation of the pileup surrounding the candidate variant. The haplotype information is encoded in the haplotype channel and reads are also sorted based on the local haplotype association of reads. The examples are provided to a Convolutional Neural Network (CNN) to provide genotype likelihood of homozygous to reference, heterozygous or homozygous alternate. Finally, in postprocessing step we take the output from the CNN and report the variant with a genotype and a likelihood of the genotype in Variant Call Format (VCF) file. A detailed description of the haplotagging algorithm with an example is provided in the online "methods" section.

## PacBio HiFi haplotype approximation and variant calling performance

DeepVariant haplotags long reads to improve genotyping accuracy during variant calling. We trained DeepVariant models with no haplotagging information, haplotagging information from DeepVariant-WhatsHap bam, and the approximate haplotagging method. We trained the model on six Genome-In-A-Bottle (GIAB) samples (HG001, HG002, HG004, HG005, HG006, HG007) and kept HG003 as a hold out sample, we also holdout chromosome 20 as hold out. We then compared the variant calling performance on HG003 sample at different coverages.

In the variant calling performance analysis (Fig. 2a and Supplementary Table 1), we observe that DeepVariant model that uses no haplotagging information has lower variant calling performance at all coverages. The variant calling performance for SNPs with no haplotagging information can be considered comparable (F1-score at **15x**: 0.9960, **35x**: 0.9990) to the performance of variant calling with haplotagging information (F1-score at **15x**: 0.996312, **35x**: 0.999292). However, the INDEL performance of variant calling without haplotag information (F1-score at **15x**: 0.9529, **35x**: 0.9906) trails behind the performance of variant calling with haplotag information (F1-score at **15x**: 0.9701, **35x**: 0.9945) showing the importance of using haplotag information during variant calling with PacBio HiFi long reads.

We also observe that the variant calling performance of DeepVariant that uses approximate haplotagging built within the variant calling process is comparable to the previously used WhatsHap-based pipeline. The DeepVariant model trained with haplotag information using approximate haplotagging (F1-score **15x**: 0.9963, **35x**: 0.9992) outperforms the WhatsHap-based DeepVariant model (F1-score **15x**: 0.9961, **35x**: 0.9947). For INDEL performance, we see that the performance of the model trained with approximate haplotagging (F1-score **15x**: 0.9701, **35x**: 0.9945) is comparable to the WhatsHap-based model (F1-score **15x**: 0.9705, **35x**: 0.9947). Overall, we see that the model that operates with haplotag information from approximate haplotagging has comparable or better variant calling performance than the WhatsHap-based method that requires running the entire variant calling process multiple times.

We compared the approximate haplotagging performance of DeepVariant against WhatsHap-based haplotagging accuracy on HG002 chr20. We took GIAB trio-phased variant calls and haplotagged the PacBio HiFi reads using WhatsHap and used that as the truth set for haplotype association for reads. For WhatsHap-based haplotagging we used DeepVariant to initially identify variants from unphased bam and then used WhatsHap haplotag to assign haplotags to reads. For approximate haplotagging with DeepVariant, we reported the read haplotype association of each chunk and merged them. Finally, we compared the read haplotype association accuracy against GIAB-based haplotagging.

Figure 2b shows DeepVariant approximate haplotagging accuracy of 99.14 is comparable to WhatsHap-based haplotagging accuracy of 99.26. Notably, DeepVariant-based haplotagging is performed on 25kbp chunks and merged at the end whereas WhatsHap operates on entire chromosome with high-quality variants from DeepVariant. We

also observe the haplotagging partitions are comparably consistent between WhatsHap-based haplotagging and DeepVariant approximate haplotagging. Overall, the approximate haplotagging method shows improved variant calling accuracy compared to non-haplotagged model (Fig. 2a, b). However, we observe slightly lower INDEL accuracy compared to WhatsHap-based method but improved SNP accuracy with approximate haplotagging method. We also compared the variant calling and approximate haplotagging accuracy for Oxford Nanopore data in Supplementary Fig. 1.

## PacBio Revio vs Sequel-II variant calling performance

PacBio has announced a new sequencing instrument called Revio. Revio is high-throughput than previous Sequel-II. We analyzed the variant calling performance of DeepVariant with approximate haplotagging on PacBio Revio and Sequel-II instruments. The model is trained on a dataset consisting of sequencing data coming from both PacBio Revio and Sequel-II instruments. We trained the model on GIAB samples of HG001-HG007 while holding out HG003 for evaluation. We analyzed the performance of DeepVariant at different coverages at various coverage between 5x to 30x coverage.

In Fig. 3a, we see the read length distribution of Revio is much wider compared to Sequel-II method as the library preparation for Sequel-II used a more precise gel-based method for read size selection. We used Phred Quality Score as Quality Value (QV) to estimate the quality of the reads. The empirical QV of both Revio and Sequel-II data are comparable.

The variant calling performance of DeepVariant with Sequel-II and Revio are shown in Fig. 3b and Supplementary Table 2. Here we see the SNP calling performance of DeepVariant is comparable between Sequel-II and Revio platforms. The SNP variant calling performance (F1-scores of Sequel-II and Revio respectively, **5x**: 0.8539 vs 0.8535, **10x**: 0.9803 vs 0.9793, **15x**: 0.9963 vs 0.9960, **20x**: 0.9985 vs 0.9985, **25x**: 0.9990 vs 0.9991, **30x**: 0.9992 vs 0.9993) are very similar between two platforms at any given coverage. We also compared the variant calling performance against LongShot[22] and NanoCaller[45] in Supplementary Fig. 2.

The INDEL performance of DeepVariant at different coverages shows some difference (F1-scores of Sequel-II and Revio respectively, **5x**: 0.740798 vs 0.718782, **10x**: 0.92547 vs 0.908092, **15x**: 0.970101 vs 0.960039, **20x**: 0.984227 vs 0.978443, **25x**: 0.990044 vs 0.986717, **30x**: 0.993007 vs 0.990944). We observed the INDEL variant calling performance of DeepVariant with Sequel-II is higher compared to Revio at coverages between 5x to 25x. Whereas, at the suggested coverage of 30x, the variant calling performance with Revio is comparable to Sequel-II. The difference in the variant calling performance could be attributed to the read length distribution difference between the two datasets. However, Revio is high-throughput and the overall resources required to generate 30x data are much fewer compared to Sequel-II which makes Revio more practical for large-scale studies.

## Oxford nanopore simplex variant calling performance

Oxford Nanopore Technologies (ONT) introduced an updated and more accurate chemistry known as R10.4 which improves on homopolymer errors in nanopore reads. The error rate of ONT R9.4.1 data requires pre-processing methods PEPPER-Margin to find candidates and haplotag the reads before variant calling with DeepVariant. However, with the improved sequencing quality from ONT R10.4 chemistry and approximate haplotagging method in DeepVariant, we are now able to train a model for ONT R10.4 that can call variants directly with DeepVariant without requiring complex preprocessing of PEPPER-Margin. We trained the DeepVariant ONT model on a combined dataset of R10.4 simplex and duplex data on GIAB samples where we held out HG003 for evaluation.

In Fig. 4a and Supplementary Tables 3 and 4, we compared the variant calling performance of ONT R10.4 data with three available

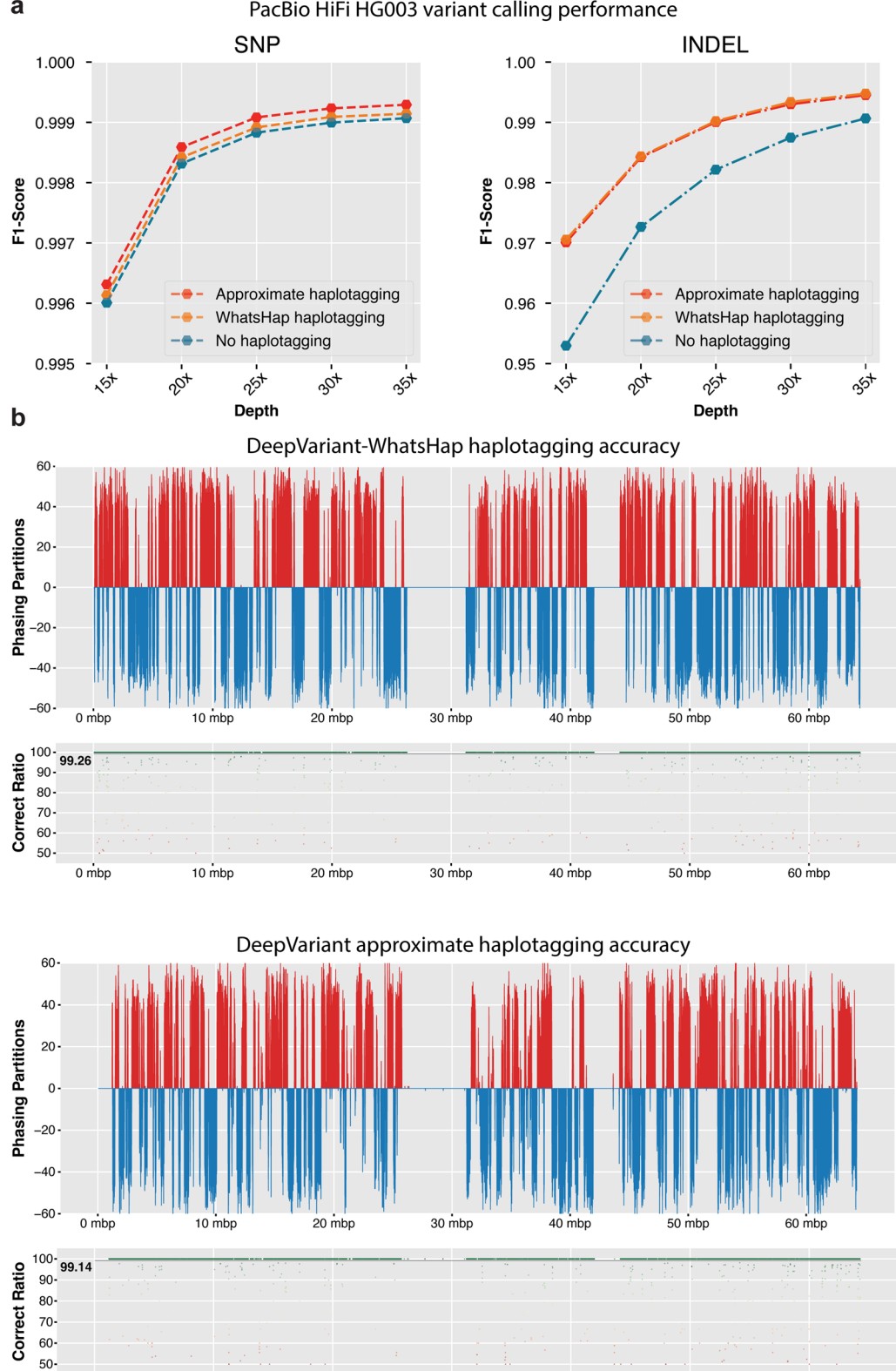

**Fig. 2 | Effectiveness of local haplotagging approximation for variant calling. a** Variant calling accuracy of PacBio HiFi reads with no haplotagging, haplotagging with WhatsHap and approximate haplotagging. **b** Haplotagging accuracy of WhatsHap and the local approximate haplotagging method.

variant callers, PEPPER, Clair3 and DeepVariant. Here, PEPPER refers to the PEPPER-Margin-DeepVariant pipeline and DeepVariant refers to DeepVariant with approximate haplotagging. In our comparison, we observe that DeepVariant with haplotagging has comparable SNP variant calling performance at all coverages (SNP F1-score, **15x:**

DeepVariant: 0.9903, PEPPER: 0.9875, Clair3: 0.9880, **30x:** DeepVariant: 0.9968, PEPPER: 0.9962, Clair3: 0.9957, **65x:** DeepVariant: 0.9976, PEPPER: 0.9977, Clair3: 0.9976). On the other hand, the INDEL variant calling performance of DeepVariant is higher compared to Clair3 and PEPPER at high and low coverages (INDEL F1-score, **15x:** DeepVariant:

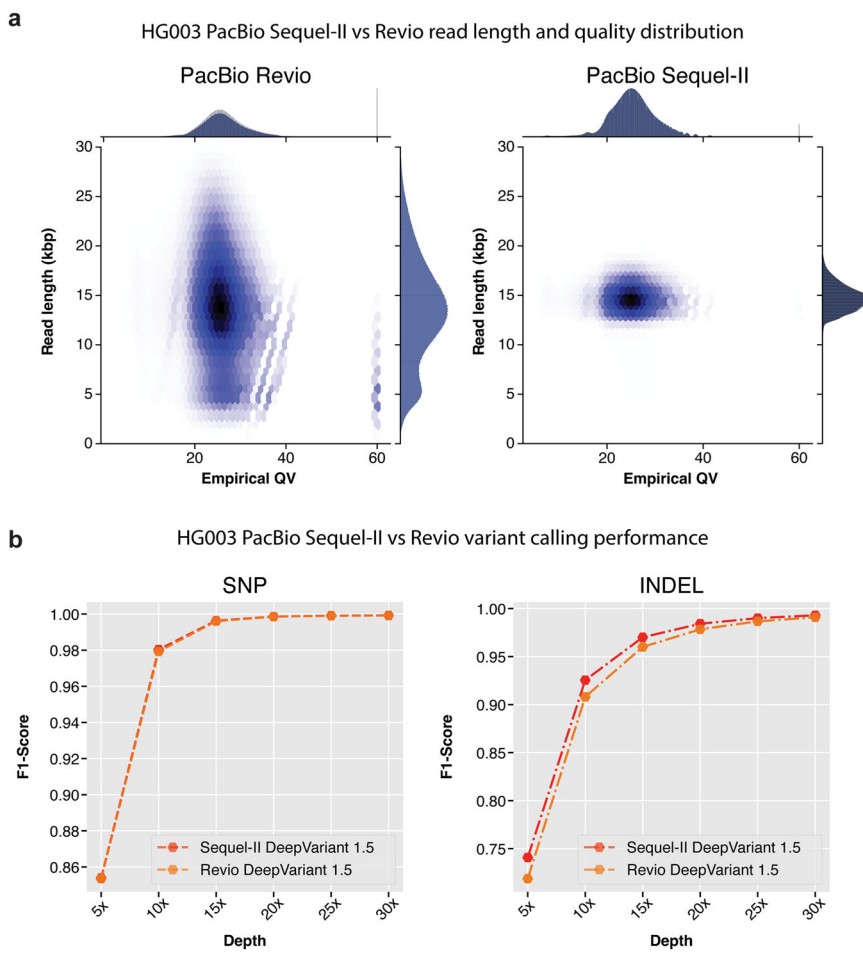

**Fig. 3 | PacBio Revio and Sequl-II variant calling performance comparison. a** Read length distribution and empirical QV distribution of reads from Revio and Sequel-II. **b** Variant calling performance of DeepVariant with Revio and Sequel-II data.

0.7121, PEPPER: 0.7013, Clair3: 0.6902, **30x**: DeepVariant: 0.8412, PEPPER: 0.8345, Clair3: 0.8400, **65x**: DeepVariant: 0.8976, PEPPER: 0.8830, Clair3: 0.8800). Overall, the DeepVariant with approximate haplotagging shows it can derive high-quality variants from ONT R10.4 sequencing data.

Figure 4b and Supplementary Tables 5 and 6 show the variant calling performance improvement between R9.4.1 and R10.4 chemistry. We observe that the SNP variant calling performance is comparable between R9.4.1 and R10.4 between 40x and 65x (SNP F1-score **40x**: R9: 0.9971, R10: 0.9969, 65x: R9: 0.9976, R10: 0.9974) coverage but R10.4 shows improvements between 10x and 35x coverage where the most improvement comes at lower coverage between 10x and 25x (SNP F1-score **10x**: R10: 0.9645, R9: 0.9371, **25x**: R10: 0.9962, R9: 0.9948). The INDEL variant calling is highly improved with R10.4 chemistry at all coverages showing the improved data quality with the updated chemistry (INDEL F1-score **15x**: R10: 0.7725, R9: 0.6666, **30x**: R10: 0.8412, R9: 0.7712, **65x**: R9: 0.8472, R10: 0.8976). Overall, the variant calling improvements we observe between R9.4.1 and R10.4 chemistry is due to the improvements in nanopore chemistry.

### Improved variant calling with Oxford nanopore duplex sequencing

With the introduction of newer R10.4 chemistry, Oxford Nanopore also announced a new data type called duplex. We assessed the variant calling performance of DeepVariant on duplex and simplex data types of R10.4 chemistry. For this analysis, we trained a model with R10.4 simplex and duplex data combined, where we had only HG002

available for duplex data. For each sample, we trained the model from chr1-chr19 and tuned on chr21-chr22 which left chr20 for evaluation. For evaluation, we compared the variant calling performance on chr20 of HG002 sample at different coverages.

Figure 5a shows the empirical QV vs read length distribution between simplex and duplex datasets. The simplex reads have a read length distribution that goes beyond 100kbp+ length reads. Although duplex read length is between 10kbp-50kbp, we observe major improvements in empirical QV with duplex data where the empirical QV reaches nearly Q30 whereas with simplex the empirical QV ranges between Q17 to Q20. With the improvements of empirical QV, duplex data is expected to deliver high-quality variant calls.

The variant calling performance comparison between simplex and duplex datatype with DeepVariant shows duplex data improves variant calling performance for both SNP and INDEL (Fig. 5b and Supplementary Table 7) at all coverages. The SNP variant calling performance at lower coverages with duplex data is improved compared to simplex data (SNP F1-score **10x**: Duplex: 0.9661, Simplex: 0.9540, **20x**: Duplex: 0.9981, Simplex: 0.9958, **30x**: Duplex: 0.9989, Simplex: 0.9981). At high coverage between 40x-65x, simplex and duplex data shows comparable SNP variant calling performance (SNP F1-score **40x**: Duplex: 0.99911, Simplex: 0.9984, **50x**: Duplex: 0.99918, Simplex: 0.9985). For INDEL variant calling, duplex shows improvements at all coverages compared to simplex data (F1-score **10x**: Duplex: 0.7432, Simplex: 0.6892, **30x**: Duplex: 0.8652, Simplex: 0.8326, **50x**: Duplex: 0.9032, Simplex: 0.8720). Overall, the duplex datatype shows improvements in variant calling in all aspects.

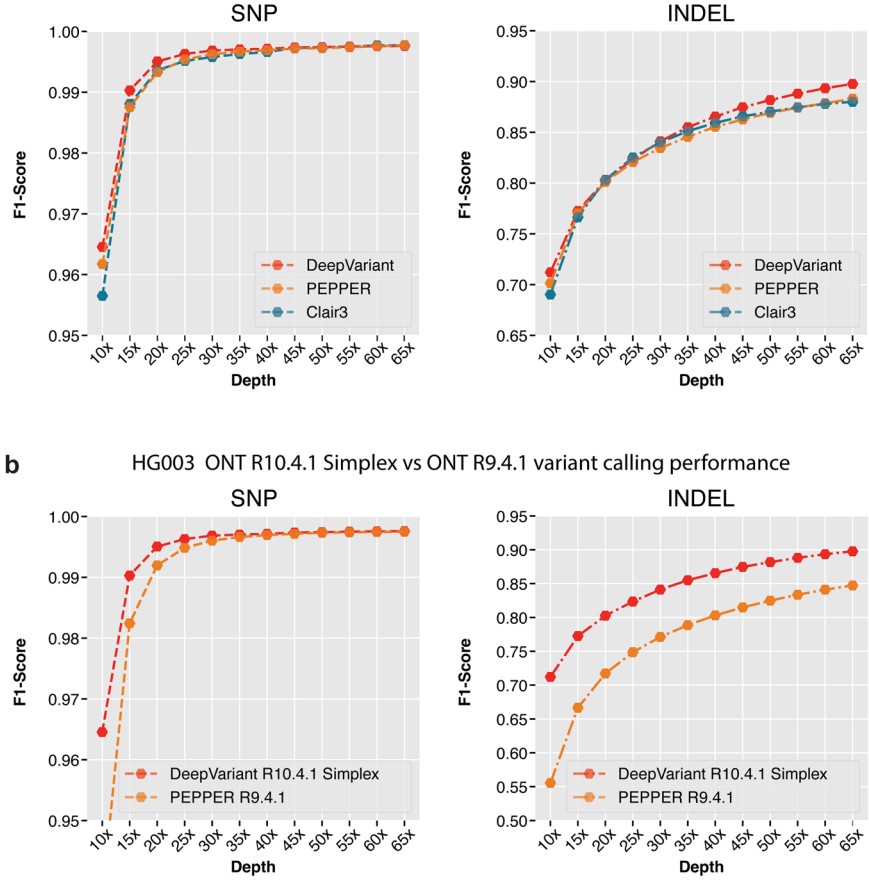

**Fig. 4 | Variant calling performance comparison between Oxford nanopore R9.4.1 and R10.4 chemistry. a** Variant calling performance of R10.4 chemistry between DeepVariant, PEPPER and Clair3 variant callers. **b** Variant calling performance of DeepVariant between R10.4 chemistry and R9.4.1 chemistry.

## Discussion

The rapid improvement in long-read sequencing technology has shown its utility from high-quality genome assembly to rare disease diagnosis[11,13,46,47]. Most variant calling methods that are tuned toward short-read sequencing technologies[48,49] fail to derive and utilize the linkage information long-reads can provide during variant calling. Currently, most long-read variant callers require haplotagging to be done by an external method like WhatsHap[31] or Margin[16] which creates a complex multi-step variant calling process[29]. Besides inference, training and iteration on these models become difficult. In this work, we introduce an approximate local haplotagging method that can haplotag reads in 25kb chunks and improves the variant calling accuracy over non-haplotagged mode with PacBio HiFi data. We also extended the support to Oxford Nanopore Technologies R10.4 chemistry showing that it is possible to directly call variants from ONT data without multi-step PEPPER-Margin-DeepVariant setup.

The approximate haplotagging method we developed works efficiently on 25kb chunks of the genome and haplotags reads with comparable accuracy to WhatsHap. We show that the approximate haplotagging method achieves 99.14% correct haplotag ratio compared to 99.26% correct haplotag ratio of WhatsHap. We also show that models trained with haplotagging information from approximate haplotagging method has comparable INDEL accuracy and higher SNP accuracy than models trained with haplotagging information with WhatsHap or with no-haplotagging information.

We show the variant calling performance on two PacBio platforms, Revio and Sequel-II. We show that the newer high-throughput Revio's variant calling performance with DeepVariant is comparable to Sequel-II platform at each coverage. At 30x coverage the performance of

Sequel-II (F1-Score INDEL: 0.993, SNP: 0.9992) is comparable to Revio (F1-Score INDEL: 0.990, SNP: 0.9993). The performance difference at lower coverages between Revio and Sequel-II is suspected to be caused by the read length distribution difference between the two datasets. We observed Sequel-II to have a tighter band around 15kb read length, whereas Revio had a wider read length distribution from 5kb to 25kb.

We extended DeepVariant to ONT R10.4 chemistry that has higher read-quality compared to R9.4.1 chemistry. We first show DeepVariant on R10.4 simplex data has the highest quality variant calls compared to PEPPER and Clair3. Then we show that DeepVariant with R10.4 is more accurate than PEPPER with R9.4.1 data. We show that with R10.4 chemistry it is possible to produce high-quality variants at lower coverages compared to R9.4.1 with demonstrable improvements in INDEL accuracy.

We also show that DeepVariant works seamlessly with R10.4 simplex and duplex data types. The duplex data has higher average read quality (average QV27) compared to simplex (average QV20) and produces even higher quality variant calls compared to simplex only data type. We show with R9.4.1 it is possible to achieve SNP F1-score of 0.999 at 30x Duplex coverage whereas 65x Simplex data achieves 0.997 SNP F1-score. For ONT INDEL performance, we see a noticeable improvement from R9.4.1 (INDEL F1-score 0.84), R10.4 simplex (INDEL F1-score 0.897) and R10.4 duplex (INDEL F1-Score 0.903). Although the INDEL improvements between platforms are on the positive side, we believe further iteration on the chemistry and data quality improvements would demonstrate further accuracy improvements in the future.

As we demonstrate a more generalized framework for long-read variant calling that uses haplotagging information, we believe future

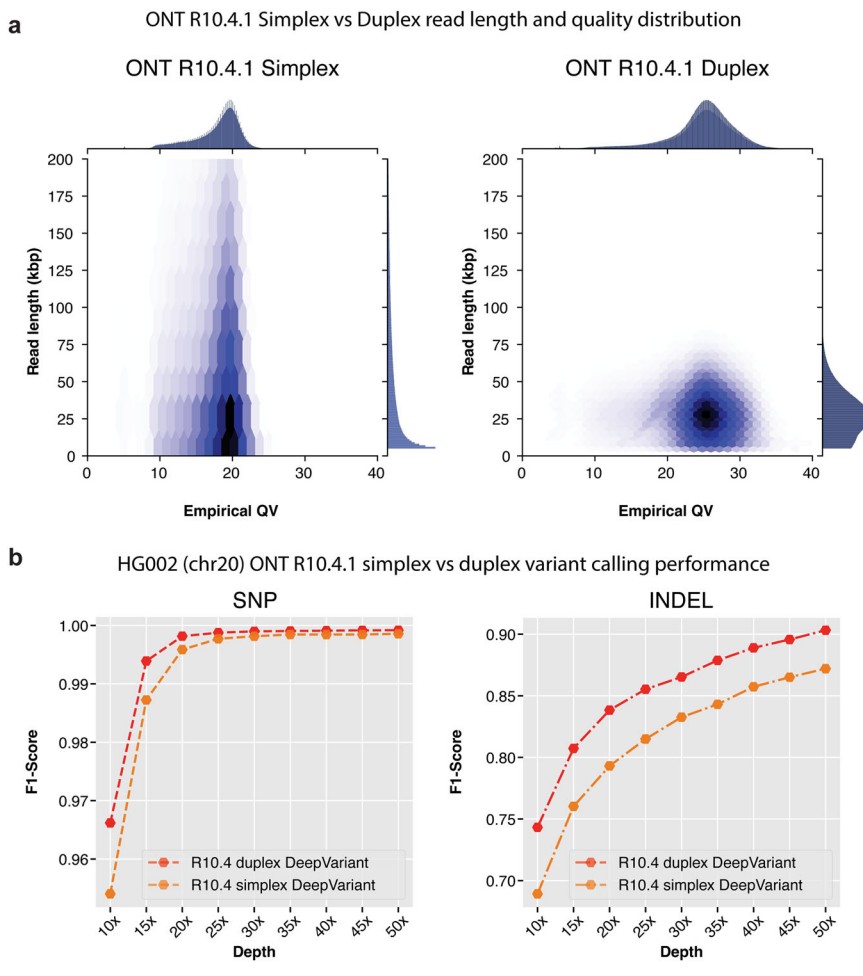

**a** ONT R10.4.1 Simplex vs Duplex read length and quality distribution

**Fig. 5 | Comparison between Oxford Nanopore simplex and duplex datatypes with R10.4 chemistry. a** Read length distribution and empirical QV distribution of reads from simplex and duplex datatypes. **b** Variant calling performance of DeepVariant with simplex and duplex datatypes.

iterations of incorporating more long-read specific features would improve the variant calling accuracy further. For example, both PacBio and ONT can now produce methylation information[50,51]. The newer methods can provide epigenetic profiles as well as canonical base calls[52,53]. We believe incorporating methylation information in the variant calling framework would further improve the accuracy.

## Methods
### Approximate haplotagging
In this section, we describe the approximate haplotagging algorithm in detail. The approximate haplotagging is applied on 25kb windows indepedently and all described methods work on the 25kb window only. We utilize a graph data structure to facilitate the haplotagging of reads. In the graph $G$, a set of vertices $V$ represents alleles at locations where a set of reads $R$ overlaps these vertices. Our objective is to assign a haplotag of 1, 2, or 0 to all vertices, ensuring that this assignment corresponds to the maximum read support. Haplotag 1 is assigned to reads overlapping a majority of the alleles of haplotype 1 in the local 25kb window, while haplotag 2 is assigned to those overlapping a majority of alleles of haplotype 2 in the same window. Haplotag 0 is designated for reads that cannot be conclusively assigned to either of the two haplotypes.

For a genomic position $n$, we look at all possible tuples of vertices $(i,j)$ from the set of vertices at this position. We use a similar approach to the score calculation for each pair at the same position while calculating scores for a pair of vertices at different variant positions. The score calculation for pair of vertices at different variant positions is described in "How scores are calculated between positions" section. The best score $S(V_{n,i}, V_{n,j})$ is calculated for each tuple where the first vertex is assigned a haplotag 1 and the second vertex is assigned a haplotag 2. The set of supporting reads $R(V_{n,i}), R(V_{n,j})$ are stored for each possible haplotag assignment. Set of reads supporting the assignment $R(V_{n,i})$ is calculated as a set of reads overlapping vertex $V_{n,i}$ and preceding vertex $V_{n-1,k}$ joined with a set of reads that overlap $V_{n,i}$ and start after position $n - 1$.

At each step of the dynamic programming algorithm, we extend the best haplotag assignment calculated for the previous position. Final assignment is calculated by backtracking from the best score for the last genomic position.

Below, we present a brief definition of variables we used to describe the approximate haplotagging algorithm:

- $V_{n,j}$ be a vertex $j$ at position $n$.
- $S(V_{n,i}, V_{n,j})$ be a score for vertices $i, j$ at genomic position $n$ so that vertex $i$ is haplotag 1 and vertex $j$ is haplotag 2. It is possible that $i$ and $j$ are the same vertex;
- $V_n$ be a set of vertices at position $n$;
- $R(V)$ be a set of reads overlapping vertex $V$;
- $R(V_1, V_2)$ be a set of reads overlapping vertices $V_1, V_2$;
- $R^*(V_{n,j})$ be a set of "new" reads that start after position $(n - 1)$ and overlap vertex $V_{n,j}$;

**Initialization.** The two steps, initialization and recursion used are described below. The first genomic position of an interval ($n = 1$), we

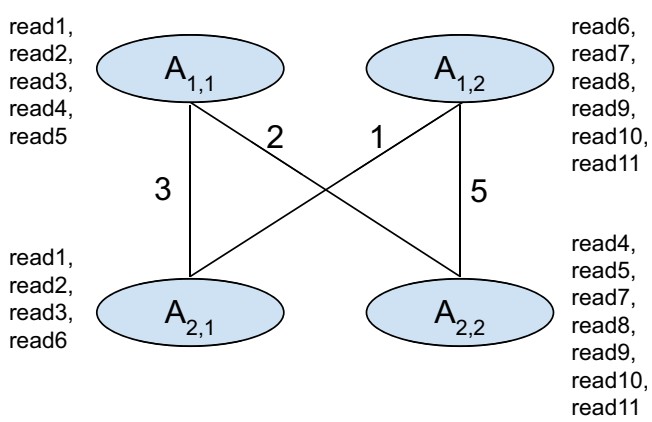

**Fig. 6 | Illustration of a graph used to explain approximate haplotagging alogrithm.** This figure demonstrates and example of how scores are calculated in the approximate haplotagging algorithm.

initialize the score as:

$$S(V_{1,i}, V_{1,j}) = size(R(V_{1,i}) \cup R(V_{1,j})) \text{ for all possible haplotagging of vertices } i \text{ and } j$$

Initialization happens at the beginning of an interval or when the haplotagging cannot be extended from the previous position.

**Recursion.** During recursion, we calculate scores at each position $n$ for all vertex pairs $(V_{n,i}, V_{n,j})$ based on the previously observed vertex $V_{n-1,k}$ and $V_{n-1,l}$ where edges $E(V_{n-1,k}, V_{n,i})$ and $E(V_{n-1,l}, V_{n,j})$ exist in the graph connecting previous position $n-1$ and current position $n$. The score is calculated as:

$$S(V_{n,i}, V_{n,j}) = max\{S(V_{n-1,k}, V_{n-1,l}) + size(R(V_{n-1,k}, V_{n,i}) \cup R(V_{n-1,l}, V_{n,j}) \cup R^*(V_{n,i}) \cup R^*(V_{n,j}))\}$$

for all pairs $(V_{n-1,k})$ and $(V_{n-1,l})$ where edges $E(V_{n-1,k}, V_{n,i})$ and $E(V_{n-1,l}, V_{n,j})$ exist in the graph.

**Backtracking and haplotag assignment.** After scores have been calculated for each vertex, we backtrack from the last position to determine the best scores, and subsequently, each pair of alleles is assigned haplotags. Now that all alleles are haplotagged, we assign reads with haplotags based on the set of haplotagged alleles the read overlaps. This assignment is accomplished by enumerating all the alleles a read overlaps and then counting how many of these alleles belong to haplotype 0, haplotype 1, and haplotype 2. Ideally, a read should overlap only alleles from the same haplotype. If a read overlaps alleles from different haplotypes then the haplotag is assigned by majority. If the number of alleles from haplotype 1 and haplotype 2 are equal, the read is assigned a haplotag 0.

**Example of the approximate haplotagging algorithm**
In Fig. 6, we illustrate a graph constructed for approximate haplotagging. In this graph each vertex represents an allele $A_{n,m}$ where $n$ is the genomic position and $m$ is the allele. The alleles ($m$) are numbered for this demonstration. Set of reads overlapping each allele is shown next to it denoted as $read1$ to $read11$. The reads are placed as such it represents the alleles it supports in the graph.

In the first step, we initialize the scores for the genomic position $n = 1$:

$$S(A_{1,1}, A_{1,1}) = size(R(A_{1,1}) \cup R(A_{1,1})) = 5$$
$$S(A_{1,1}, A_{1,2}) = size(R(A_{1,1}) \cup R(A_{1,2})) = 5 + 6 = 11$$
$$S(A_{1,2}, A_{1,2}) = size(R(A_{1,2}) \cup R(A_{1,2})) = 6$$

Then we calculate the scores of each pair of vertices based on the previously observed vertices:

$$S(A_{2,1}, A_{2,1}) = max\{S(A_{1,1}, A_{1,1}) + size(R(A_{1,1}, A_{2,1}) \cup R(A_{1,1}, A_{2,1})), S(A_{1,1}, A_{1,2})$$
$$+ size(R(A_{1,1}, A_{2,1}) \cup R(A_{1,2}, A_{2,1}))\} = max\{5 + 3, 11 + 4\} = 15$$

$$S(A_{2,1}, A_{2,2}) = max\{S(A_{1,1}, A_{1,1}) + size(R(A_{1,1}, A_{2,1}) \cup R(A_{1,1}, A_{2,2})), S(A_{1,1}, A_{1,2})$$
$$+ size(R(A_{1,1}, A_{2,1}) \cup R(A_{1,2}, A_{2,2})), S(A_{1,2}, A_{1,2})$$
$$+ size(R(A_{1,2}, A_{2,1}) \cup R(A_{1,2}, A_{2,2}))\}$$
$$= max\{5 + 3 + 2, 11 + 3 + 5, 6 + 1 + 5\} = 19$$

$$S(A_{2,2}, A_{2,1}) = max\{S(A_{1,1}, A_{1,1}) + size(R(A_{1,1}, A_{2,1}) \cup R(A_{1,1}, A_{2,2})), S(A_{1,1}, A_{1,2})$$
$$+ size(R(A_{1,1}, A_{2,2}) \cup R(A_{1,2}, A_{2,1})), S(A_{1,2}, A_{1,2})$$
$$+ size(R(A_{1,2}, A_{2,1}) \cup R(A_{1,2}, A_{2,2}))\}$$
$$= max\{5 + 3 + 2, 11 + 2 + 1, 6 + 1 + 5\} = 14$$

$$S(A_{2,2}, A_{2,2}) = max\{S(A_{1,1}, A_{1,1}) + size(R(A_{1,1}, A_{2,1}) \cup R(A_{1,1}, A_{2,1})), S(A_{1,1}, A_{1,2})$$
$$+ size(R(A_{1,1}, A_{2,1}) \cup R(A_{1,2}, A_{2,1}))\} = max\{5 + 3, 11 + 4\} = 15$$

Then we calculate the best score at the last position. The best score for the last position is 19 where allele $A_{2,1}$ is assigned to haplotag-1 and allele $A_{2,1}$ is assigned to haplotag-2. Previous pair of alleles for this score is $A_{1,1}$ with haplotag-1 and $A_{1,2}$ with haplotag-2.

Using the best score, we assign haplotags to the vertices and haplotags to the reads. From the previous step we have alleles $A_{1,1}, A_{2,1}$ assigned to haplotag-1 and alleles $A_{2,1}, A_{2,2}$ assigned to haplotag-2. Reads: $read1, read2, read3$ overlap alleles $A_{1,1}, A_{2,1}$ both of which are haplotag-1. Therefore read1, read2, read3 are assigned haplotag-1.

Reads: $read4, read5$ overlap alleles $A_{1,1}, A_{2,2}$ are consecutively haplotag-1 and haplotag-2. In that case $read4$, and $read5$ cannot be haplotagged. Following the same logic reads: $read7, read8, read9, read10, read11$ are assigned haplotag-2, and $read6$ cannot be haplotagged.

Finally, these haplotag associations of the reads are used in the examples we generate for each candidate and the DNN model uses the information to generate accurate genotypes.

**How scores are calculated between positions**
The haplotagging algorithm iterates through all tuples of edges and stores the best score for each pair of vertices. We describe three cases with examples below.

**Simple Case (for 2 positions with two edges).** In the simple case, we have two vertices per position and only two edges connecting two positions each without overlaps like in Fig. 7. In Fig. 7, vertex $A_{1,1}$ means vertex 1 in position 1 and $A_{1,2}$ means vertex 2 in position 1. So here, we would have 4 possible tuples of edges: $(A_{1,1} - A_{2,1}, A_{1,1} - A_{2,1})$, $(A_{1,2} - A_{2,2}, A_{1,2} - A_{2,2})$, $(A_{1,1} - A_{2,1}, A_{1,2} - A_{2,2})$, $(A_{1,2} - A_{2,2}, A_{1,1} - A_{2,1})$. The score is calculated for each tuple of edges. We calculate best scores for the four pairs of vertices: $S(A_{2,1}, A_{2,1})$, $S(A_{2,2}, A_{2,2})$, $S(A_{2,1}, A_{2,2})$, $S(A_{2,2}, A_{2,1})$.

**Complex case (for 2 positions with four edges).** In the worst case with four vertices for two positions, we have edges that connect all four vertices shown in Fig. 8. In this case, we will calculate scores for all

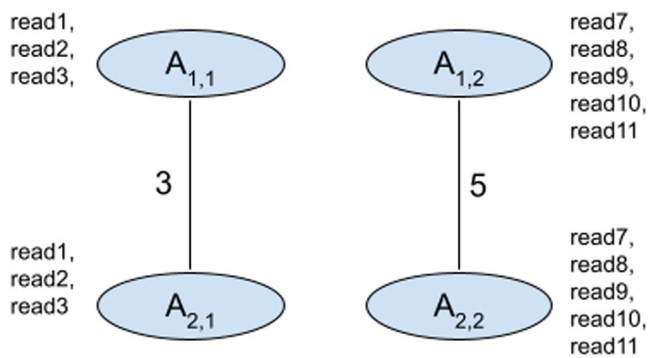

**Fig. 7 | Illustration of a graph with two vertices with two edges for simple case of score calculation.** This figure demonstrates and example of how scores are calculated in a simple case of two vertices with two edges.

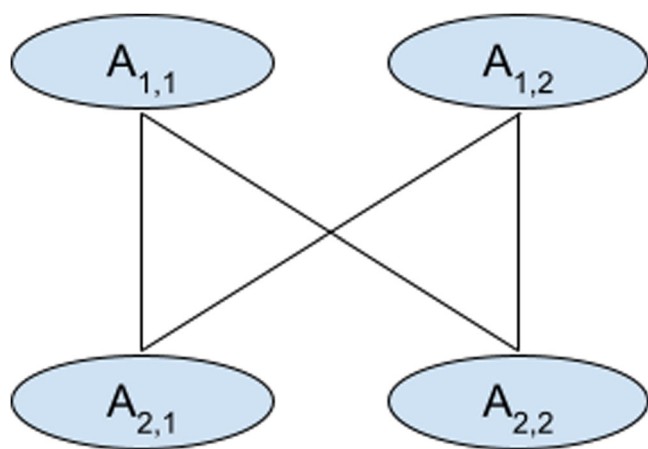

**Fig. 8 | Illustration of a graph with four vertices and four edges that connect all vertices to each other.** This figure demonstrates and example of how scores are calculated in a complex case where multiple edges require calculations of all possible combinations.

16 possible tuples of edges and we store best scores for 4 possible pairs of vertices: $S(A_{2,1}, A_{2,1})$, $S(A_{2,2}, A_{2,2})$, $S(A_{2,1}, A_{2,2})$, $S(A_{2,2}, A_{2,1})$.

**Case of 3 vertices.** In the case of three vertices, where we have reference and two alternate alleles. Similar to the previous two cases, we calculate 9 best scores for each possible pair of vertices. Depending on the number of edges, we have to calculate the scores for all possible tuples of edges to get the scores for each possible pair of vertices.

### Analysis methods
**Read alignment and subsampling.** We used `pbmm2` version `1.10` and `minimap2`[54] version `2.24-r1122` to align reads to the reference genome. We used `samtools`[55] version `1.15` for sampling alignment files at different coverages.

**Variant calling and haplotagging.** We used `PEPPER-Margin-DeepVariant`[16] version `r0.8`, `Clair3`[21] version `v1.0.0` for variant calling, for DeepVariant-WhatsHap-DeepVariant pipeline we used `v1.2.0` version of DeepVariant. For haplotagging with WhatsHap, we used WhatsHap version `v1.7`.

**Benchmarking variant calls.** For benchmarking variant calls, we used `hap.py`[56] version `v0.3.12`. The `hap.py` is available through `jmcdani20/hap.py:v0.3.12` in dockerhub. For we used GIAB v4.2.1 truth set[9] against `GRCh38` reference for all samples.

**Haplotagging accuracy and natural switch determination.** We used https://github.com/tpesout/genomics_scripts/haplotagging_stats.py to calculate the haplotagging accuracy[16].

**Read accuracy estimation.** We used `Best`[57] version `v0.1.0` for read accuracy analysis. For the analysis, we used GRCh38 as the reference to derive the empirical QV for each read.

### Reporting summary
Further information on research design is available in the Nature Portfolio Reporting Summary linked to this article.

## Data availability
We have made all data including input BAMs, output VCF and analysis files publicly available: https://console.cloud.google.com/storage/browser/brain-genomics-public/publications/kolesnikov2023_dv_haplotagging/evaluation/. Moreover, all data collected and used for this study are publicly available though the HPRC consortium: https://s3-us-west-2.amazonaws.com/human-pangenomics/index.html. The data policy and details of the data can be found in https://humanpangenome.org/data/and https://www.ncbi.nlm.nih.gov/bioproject/730823.

## Code availability
Approximate haplotagging was introduced in DeepVariant `r1.4.0` in release note: https://github.com/google/deepvariant/releases/tag/v1.4.0 and it runs by default with `run_deepvariant` without having to set any parameters explicitly to enable it for PacBio and ONT long read sequencing data. The latest versions, including `r1.5.0`, `r1.6.0` and `r1.6.1` use this feature by default similar to `r1.4.0`. Methods described here is publicly available through https://github.com/google/deepvariant where the haplotagging method can be found in https://github.com/google/deepvariant/blob/r1.6.1/deepvariant/direct_phasing.cc. The implementation of graph building method for approximate haplotagging can be found in: https://github.com/google/deepvariant/blob/r1.6.1/deepvariant/direct_phasing.cc#L481C24-L481C24. The implementation of scoring method for approximate haplotagging can be found in: https://github.com/google/deepvariant/blob/r1.6.1/deepvariant/direct_phasing.cc#L282. We have also made all input and output files publicly available. Please see the Supplementary Notes for access link.

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

## Acknowledgements

This study was supported by Google LLC (A.K., P.C., K.S., D.C., M.N., A.C.). B.P. was supported by the National Institutes of Health under award numbers R01HG010485, U24HG010262, U24HG011853, OT3HL142481, U01HG010961, and OT2OD033761. K.H.M. was supported by grants U01HG010971 and R01HG011274. We thank Trevor Pesout for the support of generating haplotag comparison plots. The content is solely the responsibility of the authors and does not necessarily represent the official views of Google LLC and the National Institutes of Health.

## Author contributions

K.S., P.C. and A.C. conceived of the study and executed it. A.K. refined and implemented an approximate haplotagging algorithm. D.C. implemented haplotype channel. M.N. implemented alignment methods that improve long-read variant calling. K.S. performed the analysis and extended it to ONT variant calling. A.C., P.C., A.K., D.C., M.N., L.B. and K.S. are core contributors and developers of DeepVariant. L.B. helped in writing the manuscript. B.M. and M.J. performed nanopore sequencing and initial analysis of the data quality. J.G. and S.G. provided analysis feedback. K.H.M., B.P. and E.A. provided guidance on the overall study and results. All authors approve of the final manuscript.

## Competing interests

A.K., P.C., K.S., D.C., M.N., A.C. and L.B. are employees of Google LLC and own Alphabet stock as part of the standard compensation package. E.A. is the founder of Personalis Inc. and Deepcell Inc., advisor to Pacific Biosciences, SequenceBio. E.A. has received support from Illumina, Oxford Nanopore, and Pacific Biosciences. Stockholder Pacific Biosciences, Oxford Nanopore. K.H.M. is a science advisory board member of Centaura; K.H.M. has received travel funds to speak at events hosted by Oxford Nanopore Technologies. J.G. holds stock in ONT and PacBio. J.G., K.S., and S.G. have accepted bursaries to attend and speak at conferences on behalf of ONT. The remaining authors declare no competing interests.
