## [Peer Review File · Nature Communications]

Local read haplotagging enables accurate long-read small variant callingREVIEWER COMMENTS

Reviewer #1 (Remarks to the Author):

Kolesnikov et. al. proposed to use local read haplotagging before using deepvariant to generate more accurate variant calling on long-read sequencing data. They clearly describe the dynamic programming used in read haplotagging for 25kb regions, and the difference between their proposed method and the three-stage deepvariant processing for variant calling. They evaluated their method on both Pacbio and Nanopore sequencing data. However, several issues below raise my concerns.

1. One of their main claims is that their proposed method makes DeepVariant a universal variant calling solution, since other methods are difficult to be extended to new data types and platforms. However, they do not explicitly show how their method achieves good performance when trained with old data types or different platforms. They also need to show the performance of two-step variant calling with DeepVariant and one-step variant calling with deepvariant after approximate haplotagging on all evaluation sections with different instruments/platforms.

2. It seems that "the phasing partitions are comparably consistent between WhatsHap-based haplotagging and DeepVariant approximate haplotagging" with slightly lower performance. But why does the approximate haplotagging method achieves variant calling accuracy than WhatsHap-based haplotagging? This indicates that inaccurate haplotagging seems to perform better and does not make sense to me. A clear explanation is thus necessary.

3. For all the comparison, Medaka, longshot and NanoCaller are not included.

4. "PacBio Revio vs Sequel-II variant calling performance": which instruments were used to generate training data, Revio or Sequel-II? "Oxford nanopore simplex variant calling performance": which chemistry were used to generate training data, R9.4.1 or R10.4.1? Without the detail, it is hard to see whether their method is a universal variant calling solution.

5. Typos need to be corrected: "Eachexample" in Page 2, "ntroduces" in Page 4, Figure 2, b is shown in the figure, whatshap or WhatsHap should be used rather than both, ", ," in Page 9 and so on.

Reviewer #2 (Remarks to the Author):

In the manuscript, the authors propose a novel approach to exploit haplotyping information for improving variant calling from long reads. They propose to use local haplotyping information, that is an approximation of the haplotagging computed via dynamic programming. In such a way, the new pipeline is able to perform variant calling without needing to call variants two times. Indeed, current pipelines require to (i) call variants without haplotyping information, (ii) perform haplotagging, and (iii) re-call variants using haplotyping information. The new method is able to skip the first round of variant calling speeding up the entire pipeline and also making it easier to maintain and extend to more sequencing technologies (since there is no dependency on external haplotagging tools). Remarkably, the new pipeline works with both ONT and PacBio HiFi data while current state-of-the-art pipelines usually work best with only one kind of data. The new haplotagging method has been included in DeepVariant and an extensive experimental evaluation proves its accuracy on 4 different sequencing technologies (ONT R9.4.1, ONT R10.4, PacBio Revio, and PacBio Sequel II). Remarkably, the new haplotagging improves variant calling accuracy over state-of-the-art.

The manuscript is interesting but it seems a bit rushed. I have some concerns on method presentation and on the formal description of the dynamic programming (online methods). Moreover, I fell like experiments are not complete. I believe the manuscript needs an extensive revision and proofreading by senior authors.

Major

The idea behind the method is clear but the presentation can be improved. Both "Local haplotype approximation method for genotyping with DeepVariant" and "Approximate haplotagging" sections

need a revision.

Here some guidelines on how to improve the two sections:

- * From the text it is not clear how a "genomic position with a putative heterozygous variant" is chosen. From Figure 1 it seems that the loci are identified from the make_examples output but I would add this to the text too. Therefore, the make_examples step is run twice, once on the reads and then another time on the haplotagged chunks? I would make this clearer.
- * It is not clear what a "partition" is.
- * It is not clear to me what happen when there is a variation with two alternate alleles (i.e., with genotype 1/2). Will there be three vertices (reference and 2 alternates) or just the two alternates?
- * The 25kb chunks are introduced at the end of the section but I found it quite confusing. At that point it is not clear how this chunking affects the method and when it is performed.
- * This also creates doubts on how the graph is built: is it a global graph containing all heterozygous variants or a local graph containing only variations falling in some sort of (25kb) window?
- * "reads are also sorted based on the haplotype association of reads", I would add "local haplotype association" since multiple tags can be assigned to a single read (since it is chunked)
- * It is not clear why you need to add artificial edges to those vertices with no incoming edges since you do not consider those vertices in the scoring. Then, after that you say "If there are no incoming edges for all vertices the phasing cannot be extended": since I am assuming that you check this before adding the artificial edges, I would move this sentence before the previous one.

Online methods:

- * I would explain again what haplotag 0, 1, and 2 means. I would also try to be consistent along the manuscript: in the main manuscript you use "haplotype-1/2" and "non-haplotagged" and later in the online methods you use "phase-1/2".
- * "For a genomic position n , if we have two vertices i and j where i is assigned to phase-1 and j is assigned to phase-2, we define the best score as $S(V_{n,i}, V_{n,j})$ ". Is the information on the phase important to define the score?
- * "for all possible pair of vertices i and j " - somehow related to previous point on 1|2 variations - is it correct to say that in the best case we have a 3 pairs (from two vertices) and in the worst case we have 6 pairs? I would make this clearer
- * is count the same as size in the Initialization equation?
- * in the recursion, is the max over all pairs (k,l) at position $n-1$? Moreover, R is defined for a single vertex but then it is used for a pair of vertices (is it a typo for S or it's some sort of intersection?). R^* is never defined.
- * "reads are assigned haplotags based on a set", which set? Is it chosen randomly among the best sets?
- * From the example, it seems that all reads overlapping alleles with different phase are tagged as 0. Is this done by considering each 25kb chunk independently? I would make this clearer here and also in the main manuscript.

Results are very interesting but they do not fully satisfy my curiosity. I believe that additional results are necessary to complete the analysis and potentially not leave any open question in the mind of the reader.

- * The comparison between haplotagging and approximate haplotagging is performed only on PacBio data. Why? It would be interesting to see the same results (as in Figure 2) on ONT data (by comparing approximate haplotagging with margin).
- * Why there is no results on "no haplotagged" reads on ONT data (Figure 4a)? I would expect a considerable gain that is worth to show
- * It is not clear to me why in Figure 4b you compared DeepVariant on R10.4.1 Simplex and PEPPER on R9.4.1 and you did not include the new DeepVariant on the older chemistry. I do not find these results meaningful enough since it is not clear if the improvements are thanks to the new chemistry or the new approximate haplotagging. The only reason I see is that training is quite expensive but then at the end of page 9, you say "For this analysis we trained a model with R9.4 and R10.4". I'm assuming

that this is a typo.

Minor

- * Figure 1: I would suggest to shuffle the input alignments to make it clearer that the reads are not haplotagged: the alignments with green and blue alleles so well separated reminded me of the split you can obtain in IGV using the "group alignments by tag". Moreover, some of the text is unreadable even at 150%-200% zoom on A4 paper. Please make it larger, and also remove the whitespace at the bottom of the figure.
- * page 4: in the first itemization, you use "would" only in the 2nd and 3rd point and not in the 1st. Is there any reason for this?
- * page 4: I would move the links to the source code from the main manuscript to the "Code Availability" section
- * Figure 2:
 - * there is no label b) in the figure
 - * What is the difference between direct haplotagging (used in the figure) and approximate haplotagging (used in the caption)? I assume they are the same and I suggest to use only approximate haplotagging. I can find only one occurrence of direct haplotagging (in the Discussion section). I would change it there too.
- * page 7: QV (that I assume to be Quality Values) is never introduced. I suggest to write it explicitly the first time.
- * page 11: the duplex SNP F1-score at 50x (0.9991) is lower than that at 40x (0.99911). Please add the missing digit (it should be 0.99918)
- * page 12: "Set of reads supporting the assignment $R(V_n, i)$ is calculated" -> "Set $R(V_n, i)$ of reads supporting the assignment is calculated", otherwise it's unclear if $R(V_n, i)$ is the set of reads or the assignment

Typos

- * abstract: "Oxford nanopore technologies", please use uppercase as done in the rest of the manuscript
- * page 2: sunch -> such
- * page 2: Eachexample -> Each example
- * page 4: "run DeepVariant ont non-haplotagged" -> "run DeepVariant on non-haplotagged"
- * page 4: "ntroduces" -> "introduces"
- * page 4: "or ref" -> "or reference allele"
- * page 4: make examples step of DeepVariant -> please use texttt as in page 2 for make_examples
- * page 6: sentence "We see the SNP performance at each coverage, model trained with haplotag information from approximate haplotagging (...) outperforms the WhatsHap-based model (...)." does not sound correct to me
- * page 6: "chromosme" -> "chromosome"
- * page 6: "Supplementary table 1" -> "Supplementary Table 1"
- * page 6: "15x:0.9963" -> "15x: 0.9963"
- * page 6: "aproximate" -> "approximate"
- * page 6: several occurrences of "whatshap" instead of "WhatsHap" (as in the rest of the manuscript)
- * page 6: "chrosome" -> "chromosome"
- * page 7: "In figure 3a" -> "In Figure 3a"
- * page 9: "and and"
- * page 9: "supplementary table" -> "Supplementary Table" (2 times)
- * page 9: "referes" -> "refers"
- * page 9: "R10:0.9969, ," -> "R10: 0.9969,"
- * page 9: "R10: 0.9645, ," -> "R10: 0.9645,"
- * page 9: "0.9948) ." -> "0.9948)."
- * page 9: "R9.4" -> "R9.4.1"
- * page 10: "supplementary table" -> "Supplementary Table"
- * page 11: "it's utility" -> "its utility"

- * page 11: "R10" -> "R10.4" (this is used in the entire manuscript)
- * page 11: "R9" and "R9.4" -> "R9.4.1"
- * page 13: "figure 6" -> "Figure 6"

REVIEWER COMMENTS

Reviewer #1 (Remarks to the Author):

Kolesnikov et. al. proposed to use local read haplotagging before using deepvariant to generate more accurate variant calling on long-read sequencing data. They clearly describe the dynamic programming used in read haplotagging for 25kb regions, and the difference between their proposed method and the three-stage deepvariant processing for variant calling. They evaluated their method on both Pacbio and Nanopore sequencing data. However, several issues below raise my concerns.

1. One of their main claims is that their proposed method makes DeepVariant a universal variant calling solution, since other methods are difficult to be extended to new data types and platforms. However, they do not explicitly show how their method achieves good performance when trained with old data types or different platforms. They also need to show the performance of two-step variant calling with DeepVariant and one-step variant calling with deepvariant after approximate haplotagging on all evaluation sections with different instruments/platforms.

- We agree with you that the word “universal” was not given proper context in this manuscript and it needed a more clarifying description. In our previous work, we showed DeepVariant can call variants with short reads (Poplin et al 2018), PacBio HiFi (Wenger et al, 2019) and finally Oxford Nanopore Technologies (ONT) (Shafin et al 2021). However, for long-read platforms, PacBio HiFi-DeepVariant relied on an external method WhatsHap for haplotagging. The other more noisy long-read platform ONT required a probabilistic candidate finder PEPPER and haplotagging method Margin to work upstream of DeepVariant to accurately call variants. So, DeepVariant was able to call variants across platforms but required external methods to pre-process the input data. In this work, we developed an internal haplotagging method that eliminates the need for WhatsHap for PacBio HiFi and PEPPER-Margin for ONT making DeepVariant suitable for short read and long read platforms. Given the issues raised, we have removed the mention of “universal” from the manuscript as it relies on previous work for context and switched to a more generalized description of the method. The following lines were changed in the manuscript:

- In the abstract the line: “ This addition of local haplotype approximation makes DeepVariant a universal variant calling solution for long-read sequencing platforms.” has been changed to “This addition of local haplotype approximation simplifies long-read variant calling with DeepVariant.”
- In the main text, the line: “and a simpler and universal variant calling approach that can enable accurate variant calling across different platforms is desirable” has been changed to “and a simpler variant calling approach that can enable accurate variant calling across different

platforms is desirable”.

2. It seems that “the phasing partitions are comparably consistent between WhatsHap-based haplotagging and DeepVariant approximate haplotagging” with slightly lower performance. But why does the approximate haplotagging method achieves variant calling accuracy than WhatsHap-based haplotagging? This indicates that inaccurate haplotagging seems to perform better and does not make sense to me. A clear explanation is thus necessary.

- Thank you for observing the inconsistency in describing the result in this section. In the sentence “*Overall, the approximate haplotagging method can improves variant calling accuracy-...*”, we meant to summarize by saying the approximate haplotagging gives accuracy improvement compared to the model that does not use haplotagging information.

On the other hand, when comparing between WhatsHap-based model’s performance against approximate haplotagging, we observe the SNP performance of approximate haplotagging is slightly higher compared to the WhatsHap-based model. However, the **INDEL performance** with approximate is **lower** than the WhatsHap-based model. Which can be attributed to the fact that WhatsHap uses high-quality input variants from DeepVariant while haplotagging, whereas, approximate haplotagging only uses candidate variants. Which is clarified in the following paragraph. We would also like to point out that the haplotagging accuracy between these two methods are very close 99.14 vs 99.26 and the truth data for this analysis (Shafin et al 2021) was generated using WhatsHap. To further clarify the results, we changed the summary line to: “*Overall, the approximate haplotagging method shows improved variant calling accuracy compared to non-haplotagged model. However, we observe slightly lower INDEL accuracy but improved SNP accuracy with approximate haplotagging method.*”

3. For all the comparison, Medaka, longshot and NanoCaller are not included.

- Thank you for suggesting this. We have run longshot and nanocaller for all three data types and presented the results. Unfortunately, we ran Medaka for 3 days and it didn’t finish. Given the long runtime and only applicable ONT (Medaka is developed by ONT), we were unable to present the results for Medaka variant caller. I We have added the following figure as a supplementary figure:

4. “PacBio Revio vs Sequel-II variant calling performance”: which instruments were used to generate training data, Revio or Sequel-II? “Oxford nanopore simplex variant calling performance”: which chemistry were used to generate training data, R9.4.1 or R10.4.1? Without the detail, it is hard to see whether their method is a universal variant calling solution.

- Thank you, the details on training parameters provided was not sufficient. For this experiment, we first trained models separately for Revio and Sequel-II and then trained a model with all data combined and saw nearly no difference in performance of the model in these two platforms. We observed the same trend for Nanopore Simplex and duplex data. So we used the combined dataset for training PacBio and ONT models. To clarify the training parameters, we have updated the ted with “*The model is trained on a dataset consisting of sequencing data coming from both PacBio Revio and Sequel-II instruments. We trained the*

model on GIAB samples of HG001-HG007 while holding out HG003 for evaluation.” for PacBio and changed “We trained the DeepVariant ONT model on a combined dataset of R10.4 simplex and duplex data on GIAB samples where we held out HG003 for evaluation.” for ONT.

5. Typos need to be corrected: “Eachexample” in Page 2, “ntroduces” in Page 4, Figure 2, b is shown in the figure, whatshap or WhatsHap should be used rather than both, “, ,” in Page 9 and so on.
 - Thank you, we have updated the manuscript to fix the typos noticed here and mentioned by the other reviewer.

Reviewer #2 (Remarks to the Author):

In the manuscript, the authors propose a novel approach to exploit haplotyping information for improving variant calling from long reads. They propose to use local haplotyping information, that is an approximation of the haplotagging computed via dynamic programming. In such a way, the new pipeline is able to perform variant calling without needing to call variants two times. Indeed, current pipelines require to (i) call variants without haplotyping information, (ii) perform haplotagging, and (iii) re-call variants using haplotyping information. The new method is able to skip the first round of variant calling speeding up the entire pipeline and also making it easier to maintain and extend to more sequencing technologies (since there is no dependency on external haplotagging tools). Remarkably, the new pipeline works with both ONT and PacBio HiFi data while current state-of-the-art pipelines usually work best with only one kind of data. The new haplotagging method has been included in DeepVariant and an extensive experimental evaluation proves its accuracy on 4 different sequencing technologies (ONT R9.4.1, ONT R10.4, PacBio Revio, and PacBio Sequel II). Remarkably, the new haplotagging improves variant calling accuracy over state-of-the-art.

The manuscript is interesting but it seems a bit rushed. I have some concerns on method presentation and on the formal description of the dynamic programming (online methods). Moreover, I feel like experiments are not complete. I believe the manuscript needs an extensive revision and proofreading by senior authors.

Major

The idea behind the method is clear but the presentation can be improved. Both "Local haplotype approximation method for genotyping with DeepVariant" and "Approximate haplotagging" sections need a revision.

Here some guidelines on how to improve the two sections:

- From the text it is not clear how a "genomic position with a putative heterozygous variant" is chosen. From Figure 1 it seems that the loci are identified from the

make_examples output but I would add this to the text too. Therefore, the make_examples step is run twice, once on the reads and then another time on the haplotagged chunks? I would make this clearer.

- Thank you very much for this feedback. We updated the text explaining how “a genomic position with a putative heterozygous variant” is derived from reads aligned to the reference as the step of the make_examples component.

To clarify the selection process, we have added the following statements (in bold) to the paragraph that initially explains the haplotagging algorithm:

*“DeepVariant processes input by dividing it into 25,000-base pair windows and processes the windows in parallel to create examples. The haplotagging algorithm is integrated into the make_examples step, utilizing raw candidate data to calculate local haplotagging for each 25,000-base pair window. For each window, DeepVariant generates candidate variants by identifying positions that differ from the reference genome. **The haplotagging process ensures that only potential heterozygous SNP candidates are selected. This involves excluding all candidates that have only one alternate allele, and the reference allele is supported by fewer than three reads. Additionally, any candidates featuring alternate alleles that are not SNPs are also eliminated.** Reads are then dynamically updated with haplotagging information. The haplotagging algorithm employs dynamic programming to determine the best haplotagging score for each possible haplotag assignment at each position within the window. A set of reads that overlap both the previous putative heterozygous variant location and the target location are used to calculate this score. The optimal allele haplotagging is determined by backtracking from the best score for the last variant position. After all alleles are assigned haplotags, read haplotags are assigned based on the majority of alleles that the read overlaps.”*

- It is not clear what a "partition" is. (double-check to see if partition is removed)
 - Thank you for drawing attention to this. We have removed this phrase as it is not clear. For context, sometimes we refer to putative heterozygous candidates as a “partition”.
- It is not clear to me what happen when there is a variation with two alternate alleles (i.e., with genotype 1/2). Will there be three vertices (reference and 2 alternates) or just the two alternates?
 - In the paragraph starting with “We use a graph to store read support information where each vertex...” we explained how vertices are created for each alt allele and ref allele. It is possible that more than two vertices are created for each alternate allele and reference allele if there is enough evidence supporting those alleles. The paragraph is updated with the following statement: “It is possible to create more than two vertices at some positions if there is substantial support for both the reference and multiple alternate alleles.”

- The 25kb chunks are introduced at the end of the section but I found it quite confusing. At that point it is not clear how this chunking affects the method and when it is performed.
 - In the Results section in the paragraph starting with “*DeepVariant processes input by dividing it into 25,000-base pair windows ..*” We explain that the algorithm works on a 25,000-base window at a time and introduce it at the very beginning of the description. Phasing is performed in a context of a 25,000 base window. We mentioned the 25kb chunk size throughout the section to maintain the context.
- This also creates doubts on how the graph is built: is it a global graph containing all heterozygous variants or a local graph containing only variations falling in some sort of (25kb) window?
 - In the Results section, we have rewritten the paragraph starting with “*We use a graph to store read support information where each vertex..*”. The paragraph now states: “*This graph is constructed from candidates identified within a 25,000-base pair window.*” which we hope clarifies how the graph is built.
- "reads are also sorted based on the haplotype association of reads", I would add "local haplotype association" since multiple tags can be assigned to a single read (since it is chunked)
 - Thank you, we have applied the edit you suggested.
- It is not clear why you need to add artificial edges to those vertices with no incoming edges since you do not consider those vertices in the scoring. Then, after that you say "If there are no incoming edges for all vertices the phasing cannot be extended": since I am assuming that you check this before adding the artificial edges, I would move this sentence before the previous one.
 - Suggested edit was applied in the paragraph starting with “*We assume that the score representing each possible phasing at a given position...*”

Online methods:

- I would explain again what haplotag 0, 1, and 2 means. I would also try to be consistent along the manuscript: in the main manuscript you use "haplotype-1/2" and "non-haplotagged" and later in the online methods you use "phase-1/2".
 - We added a clear definition of haplotag values in the paragraph starting with “*In this section, we describe the approximate haplotagging..*”. We have added: “*Haplotag 1 is assigned to reads overlapping a majority of the alleles of haplotype 1 in the local 25kb window, while haplotag 2 is assigned to those overlapping a majority of alleles of haplotype 2 in the same window. Haplotag 0 is designated for reads that cannot be conclusively assigned to either of the two haplotypes.*” which

clarifies what each haplotags are. We also changed all instances of “phase” to “haplotag”.

- "For a genomic position n , if we have two vertices i and j where i is assigned to phase-1 and j is assigned to phase-2, we define the best score as $S(Vn,i, Vn,j)$ ". Is the information on the phase important to define the score?
 - In the revised paragraph starting with "For a genomic position n , we look at each possible tuple of vertices (i, j) ...". We now introduce i, j as "we look at all possible tuples of vertices (i, j) from the set of vertices at this position". In this scenario, we take all possible tuples of vertices and calculate the score for all possible tuples. We define the score as the first vertex to be in haplotag 1 and second vertex to be in haplotag 2. We updated the statement to: "The best score $S(V(n,i), V(n,j))$ is calculated for each tuple where the first vertex is assigned a haplotag 1 and the second vertex is assigned a haplotag 2."
- "for all possible pair of vertices i and j " - somehow related to previous point on 1|2 variations - is it correct to say that in the best case we have a 3 pairs (from two vertices) and in the worst case we have 6 pairs? I would make this clearer

The haplotagging algorithm iterates through all tuples of edges and stores the best score for each pair of vertices. Therefore, in the most simple case, with two edges in figure 1, we would have 4 possible tuples of edges to look at and generate scores for four pair of vertices. In the more complex case for four edges in figure Y, we may have multiple scores for the same pair of vertices.

Simple Case:

In this case, we have two vertices per position and only two edges connecting two positions each without overlaps as shown in the following figure. In this figure, vertex $A_{1,1}$ means vertex 1 in position 1 and $A_{1,2}$ means vertex 2 in position 1. In the best case, we would have 4 possible tuple of edges: $(A_{1,1} - A_{2,1}, A_{1,1} - A_{2,1})$, $(A_{1,2} - A_{2,2}, A_{1,2} - A_{2,2})$, $(A_{1,1} - A_{2,1}, A_{1,2} - A_{2,2})$, $(A_{1,2} - A_{2,2}, A_{1,1} - A_{2,1})$. The score is calculated for each tuple of edges. We calculate best scores for the four pairs of vertices: $S(A_{2,1}, A_{2,1})$, $S(A_{2,2}, A_{2,2})$, $S(A_{2,1}, A_{2,2})$,

$S(A_{2,2}, A_{2,1})$.

Figure 1

Worst case (for 2 positions with four edges):

In the worst case with four vertices for two positions, we have edges that connect all four vertices. In this case, we will calculate scores for all 16 possible tuples of edges and we store best scores for 4 possible pairs of vertices: $S(A_{2,1}, A_{2,1})$, $S(A_{2,2}, A_{2,2})$, $S(A_{2,1}, A_{2,2})$, $S(A_{2,2}, A_{2,1})$.

Case of 3 vertices:

In the case of three vertices, where we have reference and two alternate alleles. Similar

to the previous two cases, we calculate 9 best scores for each possible pair of vertices. Depending on the number of edges, we have to calculate the scores for all possible tuples of edges to get the scores for each possible pair of vertices.

We have added this text to “How scores are calculated between positions” subsection.

- is count the same as size in the Initialization equation?
 - Thank you, yes it is the size of the set, we have changed all instances of “counts” to “size”.
- in the recursion, is the max over all pairs (k,l) at position n-1? Moreover, R is defined for a single vertex but then it is used for a pair of vertices (is it a typo for S or it's some sort of intersection?). R* is never defined.
 - Thank you, we have added definitions before the “initialization” section of online methods.

At each step of the dynamic programming algorithm, we extend the best haplotag assignment calculated for the previous position. Final assignment is calculated by backtracking from the best score for the last genomic position.

- $S(V_{n,i}, V_{n,j})$ be a score for vertices i, j at genomic position n so that vertex i is haplotag 1 and vertex j is haplotag 2. It is possible that i and j are the same vertex;
- V_n be a set of vertices at position n ;
- $R(V)$ be a set of reads overlapping vertex V ;
- $R(V_1, V_2)$ be a set of reads overlapping vertices V_1, V_2 ;
- $R^*(V_{n,j})$ be a set of “new” reads that start after position $(n - 1)$ and overlap vertex $V_{n,j}$;
- $V_{n,j}$ be a vertex j at position n .

- "reads are assigned haplotags based on a set", which set? Is it chosen randomly among the best sets?
 - In the revised paragraph starting with “After scores have been calculated for each vertex, ...” we have clarified that the reads are assigned haplotags based on the set of alleles they overlap. The section now reads as:
*“After scores have been calculated for each vertex, we backtrack from the last position to determine the best scores, and subsequently, each pair of alleles is assigned haplotags. **Now that all alleles are haplotagged, we assign reads with haplotags based on the set of haplotagged alleles the read overlaps.** This assignment is accomplished by enumerating all the alleles a read overlaps and then counting how many of these alleles belong to haplotype 0, haplotype 1, and haplotype 2. Ideally, a read should overlap only alleles from the same haplotype. If a read overlaps alleles from different haplotypes then the haplotag is assigned by majority. If the number of alleles from haplotype 1 and haplotype 2 are equal, the read is assigned a haplotag 0.”*

- From the example, it seems that all reads overlapping alleles with different phase are tagged as 0. Is this done by considering each 25kb chunk independently? I would make this clearer here and also in the main manuscript.
 - In the introduction of the approximate haplotagging method in online methods section, we added the statement: “*The approximate haplotagging is applied on 25kb windows independently and all described methods work on the 25kb window only.*” which clarifies that the method only works on the 25kb window and not whole genome.

Results are very interesting but they do not fully satisfy my curiosity. I believe that additional results are necessary to complete the analysis and potentially not leave any open question in the mind of the reader.

- The comparison between haplotagging and approximate haplotagging is performed only on PacBio data. Why? It would be interesting to see the same results (as in Figure 2) on ONT data (by comparing approximate haplotagging with margin).

→ Thank you for raising this concern. There are a few issues that make it difficult to accurately

benchmark the haplotagging accuracy of ONT data. Following are the reasons:

- 1) **Chunks vs read lengths:** In approximate haplotagging, we divide the entire chromosome into 25kb chunks and perform haplotagging only on that chunk. ONT read lengths (~100kb) are much larger than 25kb chunks so if you perform haplotagging on the entire chromosome like Margin does, the phaseblocks become much longer. However, when we chunk the chromosome into smaller parts, same reads that span multiple chunks receive multiple haplotags depending on what variants are observed in that local chunk. So relative to a global method, the local approximate haplotagging may seem less accurate. However, during variant calling, the model only sees 300bp window, so global haplotagging accuracy do not provide any accuracy boost which we observe in the variant calling performance.
- 2) **Derivation of truth set:** We derive the truth set using Margin that takes GIAB truth as input to phase the data which is very close to what Margin uses for haplotagging.
- 3) **Called variants vs candidates:** As margin works on called variants vs approximate haplotagging works only with candidates, margin has a higher accuracy in general. Because ONT has higher error rate, the candidates are more noisy than called variants used by Margin.

To get around some of these issues, we created a merge script that can re-haplotag reads that span across multiple chunks to assess a global haplotagging accuracy. However, due to the post-processing involved, the global haplotagging accuracy in ONT data lags the performance of Margin.

With PacBio, all of these issues are minimal as read length is on par with chunk size and candidates are more accurate. But with ONT, the global haplotagging accuracy does not seem to reflect the accuracy gain we observe in local haplotagging accuracy. As we could not find a way to derive, assess and plot haplotagging accuracy at 25kb chunks that we do for approximate haplotagging, we didn't perform ONT-specific analysis for figure 2.

However, on your request, we are presenting the results although they are very confounding to what variant calling accuracy we observe with approximate and Margin haplotagging.

To summarize, the haplotagging is only used for variant calling, so the phase block and global accuracy does not necessarily reflect the improvements in variant calling accuracy. Moreover, we did not find a good way to quantify and report local haplotagging accuracy hence we did not do the same analysis for longer ONT reads.³

- Why there is no results on "no haplotagged" reads on ONT data (Figure 4a)? I would expect a considerable gain that is worth to show
 → Thank you, we see significant improvement in INDEL calling with approximate haplotagging. The results are combined and presented in the previous figure.

- It is not clear to me why in Figure 4b you compared DeepVariant on R10.4.1 Simplex and PEPPER on R9.4.1 and you did not include the new DeepVariant on the older chemistry. I do not find these results meaningful enough since it is not clear if the improvements are thanks to the new chemistry or the new approximate haplotagging. The only reason I see is that training is quite expensive but then at the end of page 9, you say "For this analysis we trained a model with R9.4 and R10.4". I'm assuming that this is a typo.

→ Our intention for Figure 4b is to draw attention to the chemistry upgrade with Nanopore sequencing technology and not variant calling improvements. The R9.4.1's high error rate didn't allow us to train models directly with DeepVariant as it would generate too many candidates. So we developed PEPPER-DeepVariant where PEPPER does the candidate finding and DeepVariant calls the variants. So, the R9.4.1 PEPPER pipeline is the result of the DeepVariant model trained for R9.4.1 chemistry.

In the results section we mention "*show the variant calling performance improvement between R9.4.1 and R10.4 chemistry*". We have further added the line: "*Overall, we observe the variant calling improvements we observe between R9.4.1 and R10.4 chemistry is due to the improvements in nanopore chemistry.*" to avoid confusion.

Finally, thank you for noticing ""For this analysis we trained a model with R9.4 and R10.4", it was a typo. It was meant for simplex and duplex data. It has been changed to "*For this analysis we trained a model with R10.4 simplex and duplex data combined*" which is appropriate for that section.

Minor

- Figure 1: I would suggest to shuffle the input alignments to make it clearer that the reads are not haplotagged: the alignments with green and blue alleles so well separated reminded me of the split you can obtain in IGV using the "group alignments by tag". Moreover, some of the text is unreadable even at 150%-200% zoom on A4 paper. Please make it larger, and also remove the whitespace at the bottom of the figure. Thank you very much for raising this concern. We have re-created figure-1, we hope you find it clearer in the improved version:

- page 4: in the first itemization, you use "would" only in the 2nd and 3rd point and not in the 1st. Is there any reason for this?
 - Thank you, we have updated the text to reflect the correct grammar. We have replaced would to "is" for both occurrences.
- page 4: I would move the links to the source code from the main manuscript to the "Code Availability" section
 - Thank you, we have moved the links to the Code Availability section.
- Figure 2:
 - there is no label b) in the figure
 - What is the difference between direct haplotagging (used in the figure) and approximate haplotagging (used in the caption)? I assume they are the same and I suggest to use only approximate haplotagging. I can find only one occurrence of direct haplotagging (in the Discussion section). I would change it there too.

- Thank you, we have changed all instances of “direct haplotagging” to “approximate haplotagging”.
- page 7: QV (that I assume to be Quality Values) is never introduced. I suggest to write it explicitly the first time.
 - Thank you, we have added “*We used Phred Quality Score as Quality Value (QV) to estimate the quality of the reads.*” as the definition of QV where we first mentioned QV.
- page 11: the duplex SNP F1-score at 50x (0.9991) is lower than that at 40x (0.99911). Please add the missing digit (it should be 0.99918)
 - Thank you, we have added the missing digit.
- page 12: "Set of reads supporting the assignment $R(V_n, i)$ is calculated" -> "Set $R(V_n, i)$ of reads supporting the assignment is calculated", otherwise it's unclear if $R(V_n, i)$ is the set of reads or the assignment
 - Thank you, we have updated the text to reflect the change you suggested.

Typos

Thank you for suggesting the edits. We have updated the manuscript to reflect changes suggested.

- abstract: "Oxford nanopore technologies", please use uppercase as done in the rest of the manuscript
 - Thank you, we have edited the text in the manuscript.
- page 2: sunch -> such
 - Thank you, we have edited the text in the manuscript.
- page 2: Eachexample -> Each example
 - Thank you, we have edited the text in the manuscript.
- page 4: "run DeepVariant ont non-haplotagged" -> "run DeepVariant on non-haplotagged"
 - Thank you, we have edited the text in the manuscript.
- page 4: "ntroduces" -> "introduces"
 - Thank you, we have edited the text in the manuscript.
- page 4: "or ref" -> "or reference allele"
 - Thank you, we have edited the text in the manuscript.
- page 4: make examples step of DeepVariant -> please use texttt as in page 2 for make_examples
 - Thank you, we have edited the text in the manuscript.
- page 6: sentence "We see the SNP performance at each coverage, model trained with haplotag information from approximate haplotagging (...) outperforms the WhatsHap-based model (...)." does not sound correct to me

- We edited the text to: “The DeepVariant model trained with haplotag information using approximate haplotagging (...) outperforms the WhatsHap-based DeepVariant model (...)”.
- page 6: "chromosome" -> "chromosome"
 - Thank you, we have edited the text in the manuscript.
- page 6: "Supplementary table 1" -> "Supplementary Table 1"
 - Thank you, we have fixed all instances of this notation.
- page 6: "15x:0.9963" -> "15x: 0.9963"
 - Thank you, we have updated the text.
- page 6: "aproximate" -> "approximate"
 - Thank you, we have updated the text.
- page 6: several occurrences of "whatshap" instead of "WhatsHap" (as in the rest of the manuscript)
 - Thank you, we have fixed all instances of this.
- page 6: "chromosome" -> "chromosome"
 - Thank you, we have updated the
- page 7: "In figure 3a" -> "In Figure 3a"
 - Thank you, we have fixed all instances of this issue.
- page 9: "and and"
 - Thank you, we have updated the manuscript.
- page 9: "supplementary table" -> "Supplementary Table" (2 times)
 - Thank you, we have fixed all instances of this notation.
- page 9: "referes" -> "refers"
 - Thank you, we have updated the manuscript.
- page 9: "R10:0.9969, ," -> "R10: 0.9969,"
 - Thank you, we have updated the manuscript.
- page 9: "R10: 0.9645, ," -> "R10: 0.9645,"
 - Thank you, we have updated the manuscript.
- page 9: "0.9948)." -> "0.9948)."
 - Thank you, we have updated the manuscript.
- page 9: "R9.4" -> "R9.4.1"
 - Thank you, we have updated the manuscript.
- page 10: "supplementary table" -> "Supplementary Table"
 - Thank you, we have updated the manuscript.
- page 11: "it's utility" -> "its utility"
 - Thank you, we have updated the manuscript.
- page 11: "R10" -> "R10.4" (this is used in the entire manuscript)
 - Thank you, we have fixed all instances of this notation.
- page 11: "R9" and "R9.4" -> "R9.4.1"
 - Thank you, we have fixed all instances of this notation.
- page 13: "figure 6" -> "Figure 6"
 - Thank you, we have updated the manuscript.

REVIEWERS' COMMENTS

Reviewer #1 (Remarks to the Author):

The authors have addressed the concerns I had. I have no further comments.

Reviewer #1 (Remarks on code availability):

The code contains enough documents for installing and testing the results.

Reviewer #2 (Remarks to the Author):

I thank the authors for taking into account my comments. The revised manuscript addresses my concerns.

However, I have an additional concern on code availability section. The urls points to specific lines in the source but the code refers to release 1.5. Since I found the same methods in the latest release (r1.6.1), I would update the urls. Moreover, I am assuming that latest versions of deepvariant are using approximate haplotagging by default. If yes, I would make this clearer in the code section. If not, it is not clear to me how to use the approximate haplotagging. Is there any info in deepvariant README? I apologize if this is already described somewhere (in case, I would ask to add a link to the corresponding README).

Regarding the manuscript, I still have some minor comments:

- * Page 1: "F1-score of 0.84", score is misspelled.
- * Page 3: double ((before Figure 1a.
- * Page 4: "at each position within the window", I would say "variant position" to remark that it works on variations and not all 25k positions
- * Page 4: "position. haplotagging i", capitalize haplotagging.
- * Page 4: I would use an enumerate for the brief step-by-step description of the algorithm.
- * Figure 2: still an occurrence of "direct haplotagging", please replace with "approximate haplotagging".
- * Page 6: "The model is trained on GIAB samples from both sequencing platforms and HG003 is held out from the training. We analyzed the performance of DeepVariant at different coverages at various coverage between 5x to 30x coverage." is a repetition (it's already said in the previous red paragraph).
- * Page 9: "Supplementary Table 3, 4" -> "Supplementary Tables 3 and 4" - same for "Supplementary Table 5, 6".
- * Page 9: "Overall, we observe the variant calling improvements we observe between" - please remove the first "we observe".
- * Page 12: "How scores are calculated for pair of vertices is described in "How scores are calculated between positions" section.". I would remark that those scores are for pair of vertices at different (hence consecutive) variant positions. This is different from the DP formulation, that works for each pair of vertices at the "same position" but uses edge information for the recursive step. I would make the distinction clearer and I would also highlight better why you need the second section.
- * Page 13: please add a brief introduction to the itemization describing the variables. Move last point to first position. Use notation $V_{\{n,j\}}$ for points 3 and 4.
- * Page 15: "4 possible tuple", tuples
- * Page 15: "A_{1,2}" - 1,2 is not subscript.

Reviewer #2 (Remarks on code availability):

I have been able to test deepvariant on the example data but it is not clear if I ran the approximate

haplotagging or not. I installed via docker (r1.6.1) but code refers to specific commits from r1.5. I added a short comment on this in my review.

Reviewer #1 (Remarks to the Author):

The authors have addressed the concerns I had. I have no further comments.

- Thank you for your careful review.

Reviewer #1 (Remarks on code availability):

The code contains enough documents for installing and testing the results.

- Thank you for your careful review.

Reviewer #2 (Remarks to the Author):

I thank the authors for taking into account my comments. The revised manuscript addresses my concerns.

1. However, I have an additional concern on code availability section. The urls points to specific lines in the source but the code refers to release 1.5. Since I found the same methods in the latest release (r1.6.1), I would update the urls. Moreover, I am assuming that latest versions of deepvariant are using approximate haplotagging by default. If yes, I would make this clearer in the code section. If not, it is not clear to me how to use the approximate haplotagging. Is there any info in deepvariant README? I apologize if this is already described somewhere (in case, I would ask to add a link to the corresponding README).

- Thank you, we agree that given the algorithm runs by default, we needed to explicitly clarify that it's implemented and running without having to set any parameters. So we have added the following text to the code availability section which clarifies the implementation further:

Approximate haplotagging was introduced in DeepVariant r1.4.0 <https://github.com/google/deepvariant/releases/tag/v1.4.0> and it runs by default with `run_deepvariant` without having to set any parameters explicitly to enable it for PacBio and ONT long read sequencing data. The latest versions, including r1.5.0, r1.6.0 and r1.6.1 use this feature by default similar to r1.4.0.

We have also updated the links to the algorithm to the latest release as they haven't been changed since r1.4.0.

Regarding the manuscript, I still have some minor comments:

- Page 1: "F1-socre of 0.84", score is misspelled.

- Thank you, we have updated the text.
- Page 3: double ((before Figure 1a.
 - Thank you, we have updated the text.
- Page 4: "at each position within the window", I would say "variant position" to remark that it works on variations and not all 25k positions
 - Thank you, we have updated the text.
- Page 4: "position. haplotagging i", capitalize haplotagging.
 - Thank you, we have updated the text.
- Page 4: I would use an enumerate for the brief step-by-step description of the algorithm.
 - Thank you, we have updated the text.
- Figure 2: still an occurrence of "direct haplotagging", please replace with "approximate haplotagging".
 - Thank you, we have updated the text in the figure.
- Page 6: "The model is trained on GIAB samples from both sequencing platforms and HG003 is held out from the training. We analyzed the performance of DeepVariant at different coverages at various coverage between 5x to 30x coverage." is a repetition (it's already said in the previous red paragraph).
 - Thank you, we have updated the text.
- Page 9: "Supplementary Table 3, 4" -> "Supplementary Tables 3 and 4" - same for "Supplementary Table 5, 6".
 - Thank you, we have updated the text.
- Page 9: "Overall, we observe the variant calling improvements we observe between" - please remove the first "we observe".
 - Thank you, we have updated the text.
- Page 12: "How scores are calculated for pair of vertices is described in "How scores are calculated between positions" section.". I would remark that those scores are for pair of vertices at different (hence consecutive) variant positions. This is different from the DP formulation, that works for each pair of vertices at the "same position" but uses edge information for the recursive step. I would make the distinction clearer and I would also highlight better why you need the second section.
 - Thank you, we have updated the text to the following:
We use a similar approach to the score calculation for each pair at the same position while calculating scores for pair of vertices at different variant positions. The score calculation for a pair of vertices at different variant positions is described in "How scores are calculated between positions" section.
- Page 13: please add a brief introduction to the itemization describing the variables. Move last point to first position. Use notation $V_{\{n,j\}}$ for points 3 and 4.
 - Thank you, we have updated the text.
- Page 15: "4 possible tuple", tuples
 - Thank you, we have updated the text.
- Page 15: "A_{1,2}" - 1,2 is not subscript.
 - Thank you, we have updated the text.

Reviewer #2 (Remarks on code availability):

1. I have been able to test deepvariant on the example data but it is not clear if I ran the approximate haplotagging or not. I installed via docker (r1.6.1) but code refers to specific commits from r1.5. I added a short comment on this in my review.
 - Thank you, we have updated the code availability section that mentions that approximate haplotagging runs by default with run_deepvariant. Which hopefully clarifies the implementation.
-